# Structural Responses of a Conceptual Microsatellite Structure Incorporating Perforation Patterns to Dynamic Launch Loads

Sarmad Dawood Salman Dawood [1,2] and Mohammad Yazdi Harmin [1,*]

1   Department of Aerospace Engineering, Faculty of Engineering, Universiti Putra Malaysia, Serdang 43400, Selangor, Malaysia
2   Ministry of Science and Technology, Baghdad 10070, Iraq
*   Correspondence: myazdi@upm.edu.my

**Abstract:** Satellite systems undergo several operational phases during their service life, including the assembly phase, ground transportation phase, the launch phase, and the in-orbit operation phase. Among these phases, the one that imposes the highest level of loadings on the satellite is the launch phase. This phase involves a number of highly dynamic loads, all being imposed upon the satellite simultaneously. Investigation of the responses of the structural subsystem of a satellite to these loadings, namely its maximum deformations and maximum von Mises stresses, is critical if a reasonably high level of confidence is to be achieved. This confidence is in terms of ensuring that no material yielding develops in the structure as a result of the imposed launch loadings. In an earlier work, the structural subsystem of a conceptual microsatellite was designed, employing aluminum 6061 alloy as its material. It was then modified through introducing sets of parametrically defined geometric patterns as perforation patterns to remove material, towards reducing the structure's total mass, as an alternative to employing composite materials. That effort led to a mass reduction percentage of 23.15%. The current work's research effort focused on computing the responses of the perforated structure to three of the dynamic launch loads that are imposed upon satellites while being launched, namely quasi-static, random, and shock loads. These responses were then compared to those of the baseline, unperforated, version of the same structure. The values of these loads were taken from the relevant sources, with the values being nominal, and represented the loads that any satellite must qualify for before it can be accepted by the provider for inclusion in a launcher. After imposing these load values upon the structural design it was found that the structural responses indicated that the structure would successfully survive these loads without developing stresses that would lead to material yielding failure. This was deduced from computing the yield margins of safety for each loading case, and all margin values were positive, indicating that the structure, at its current development stage, did have sufficient capacity to withstand these loads without material yielding. This reinforced the conclusion of the earlier work, namely that the perforation concept did have sufficient merit to be further developed towards being implemented in future satellite designs.

**Keywords:** microsatellite; small satellite; mass reduction; structure; perforations; quasi-static analysis; random vibration analysis; shock analysis





## 1. Introduction

Mass reduction of any space system is a primary design driver that must be considered when developing its basic design. This is because the system's total mass will directly contribute to the cost of transporting it from the earth's surface to its operational orbit, through utilizing a suitable launcher system. The cost of launch varies in proportion to the total mass. As such, the total mass will directly contribute to the launcher selection process, as shown by Drenth et al. [1]. As an example of launch costs, Jones [2,3] demonstrates the average cost of launching one kilogram onboard the SpaceX Falcon 9, a widely utilized

launcher, to be around 2720 USD/kg. As such, a reduction of the total mass of any space system will lead to major program cost reductions.

Starting from the release of the so-called "Cubesat" standard in 1999 [4], the trend in astronautic design has been to continue the development and launching of small satellites. These systems have total masses ranging from less than 1 kg up to a maximum of 1000 kg, as described in works by Sweeting [5], Kramer and Cracknell [6], Xue et al. [7], and others. Satellites in the mass ranges starting from 10 kg up to 100 kg are classified as microsatellites, and they are generally capable of carrying payloads that can generate scientifically significant data while being efficient in terms of their power consumption. These satellites are generally employed in earth observation and remote sensing missions, in addition to data collection missions, scientific missions such as astronomy, and other non-time-dependent applications.

Due to the effect of the total system mass on the launch costs, it can be seen that reducing the mass of any new satellite is a primary design goal towards launch cost reduction. Mass reduction can be achieved through changing the configuration of the structure, through modifying the number of structural components included or selecting a suitable geometric configuration. This is considered part of a larger design process known as topology optimization. This process deals with achieving a mass reduction goal by changing the geometries of specifically selected structural components within the overall design. Lim et al. [8] and Viviani et al. [9] show examples of this process, among others.

However, mission requirements can impose constraints on changing structural components. Hence, an alternate method towards reducing mass is to change the materials included in the structural design, through implementing advanced materials, e.g., composite materials, sandwich panels, etc. A review of past works and satellite designs shows that this implementation of composite materials has been utilized for microsatellites with total masses exceeding 30 kg in general. Examples are provided in works by Cho and Ree [10], Kuo et al. [11], Zhengchun et al. [12], Kwon et al. [13], and others. These works describe the implementation of composite materials in the form of sandwich panels, by which thin plates of composite materials are bonded to hexagonally shaped cores of aluminum. These composite materials are specially selected for their properties, which reduce the so-called outgassing phenomenon. Namely, the chemical compounds present in these materials change into gaseous form when the satellite reaches its operational orbit in the vacuum of space. These materials are referred to as space-qualified composite materials.

Nevertheless, utilizing space-qualified composite materials, in terms of their design, analysis, and fabrication, requires specialized knowledge, high-cost software, and specialized fabrication tooling and procedures. Plus, the relatively high costs of the materials themselves is also a factor in increasing the overall costs of implementing these materials. Hence, an alternative method to reduce mass without the high costs inherent in implementing advanced materials was proposed in a previous work [14], and the current work is a continuation of that work. This alternative method was based upon designing and implementing patterns of parametrically defined geometric shapes. Repetitions of these shapes were developed into perforation patterns, which directly removed material from selected metallic structural components within the overall structural design, and which led to reductions in mass.

The concept of implementing perforations on metallic structural components has been utilized before in engineering applications. Works by Cunningham et al. [15], Abdelrahman et al. [16], Ghonasgi et al. [17], Jeong and Jhung [18], and others describe the effect of the presence of perforations on the natural frequencies of plates. In the civil engineering field, Formisano et al. [19] implemented perforations in shear panels, and found that choosing suitable patterns allowed structures to experience relatively large shear deflections without reducing their stiffness and ductility to large degrees. In acoustic soundproofing, Sailesh et al. [20] found that the presence of perforations had a positive effect on performance.

In terms of implementing perforations in astronautics applications, Millan et al. [21] described their utilization in the structures of small satellites since their introduction in 1999. Literature indicates that they have been implemented in the structures of satellites with total masses falling within the nanosatellite mass classification, with mass values below 30 kg. A well-known application of this can be clearly seen in the structure of the ISO standard CubeSat design cited previously, as set by the California Polytechnic State University (Cal-poly) [22]. However, perforations have not been generally implemented in microsatellite and larger mass classifications, and space-qualified composite materials have been implemented for structural mass reduction instead, as shown in works by Jin et al. [23], Wagih et al. [24], Ontaç et al. [25], Salem et al. [26], Slimane et al. [27], and many others.

In the earlier work, the authors described methodologies that were developed towards designing the structural subsystem for a conceptual small satellite, with a nominal total mass of less than 100 kg, including all subsystem components, hence falling within the microsatellite mass classification range. A baseline design of the structure, which was sized to meet the dimensional requirements of a number of currently active launch systems in terms of its outer dimensions, was developed, and was designed to be fabricated from aluminum 6061 alloy. Subsequently, the total mass of this baseline design was reduced through perforating a number of its structural components with parametrically defined geometric patterns, instead of implementing composite materials, as previously mentioned.

Modal analysis, specifically comparisons of the structure's fundamental natural frequency between the perforated cases and the baseline case, was utilized as a design tool towards selecting the best geometric pattern alternative from a design space. This was performed in contrast to utilizing it as a downstream analysis and verification process, as it was employed in many other works, such as Wei et al. [28], Liu et al. [29], and many others. A computational approach to modal analysis was taken, due to the large monetary requirements that would have been needed to implement an experimental approach to generate the result sets. However, experimental modal analysis (EMA) was implemented in the research effort towards analyzing certain structural components from the research effort's structural design, for validating the computational modal analysis approach. EMA has been implemented in numerous previous works, such as Rosly et al. [30], Othman et al. [31], Nadkarni et al. [32], and many others. Through EMA, the computationally computed modal results were successfully validated, hence affording a reasonable level of confidence in the computational results.

The next step involved designing a design space of geometric patterns, that were hence employed as perforation patterns. This design space included parametrically defining four shapes, namely diamonds, hexagons, squares, and triangles, and five modifications to each of these patterns, including varying both scale factors and aspect ratios, for a total of twenty alternatives. Implementing the selected best perforation pattern, after finalizing it to refine its modal characteristics, resulted in a mass reduction percentage of 23.15%, with a fundamental natural frequency that matched that of the baseline, unperforated, design.

The same work also included a preliminary investigation of the finalized perforated structure's response to one of the nominal dynamic load cases that were expected to be imposed upon the structure. Namely, quasi-static loadings applied along the structure's longitudinal and lateral directions, to determine its structural responses to these loadings, namely its maximum deformations and maximum von Mises stresses. The same response values were also computed for the baseline case, and the two result sets were compared, to investigate the effect of introducing the perforation patterns on the structural responses to the quasi-static loadings.

The current work will expand upon the investigation of the effect of the presence of the perforation patterns on the structural responses of the baseline and finalized perforated cases. This was achieved through taking not only the quasi-static loadings into consideration, but also two other loading types, namely the random vibration and shock loadings. The conclusions of the current work provided the definitive answer to the ques-

tion of whether the presence of the perforation patterns was beneficial to the structural performance of the structure, or if their presence was detrimental to that performance.

Before describing the three launch loading types that were considered in the current work, a short description of the phases any satellite system undergoes within its operational lifetime was presented. This was to provide a perspective on the role launch loads play in the overall loading environment, and the reason the current work focuses exclusively on the launch loads. Satellite systems operate when placed into orbits around the earth, at various altitudes. However, they start their operational lifetimes at assembly and testing facilities, and they go through a number of phases until they reach their final orbital positions and start operations, based upon their specific missions. A number of sources, including Wertz et al. [33], Sarafin [34], Wijker [35], Fortesque [36], and others, describe the operational phases of a satellite's life, which can be divided into three main phases:

1.  Pre-launch phase: This phase starts at the satellite's assembly facility, continues at the test facilities that clear it for integration onto the launcher and to be transported to the launch facility (by land, sea, or air transport), and ends when the satellite is mated to the launch vehicle in preparation for launch.

2.  Launch phase: This phase typically lasts for approximately eight minutes, which is the time the launcher needs to propel itself and its satellite payloads from the launch site on the earth's surface up to orbital altitudes. However, during these minutes, the satellite will experience the most intense dynamic loading it will ever experience during its full operational life, since during this phase it will be mechanically mated to the launcher. The launcher will convey to the satellite, through the launcher–satellite interface, the loading effects of multiple sources of dynamic excitation that all act on the satellite at a number of stages of the launch process.

3.  In-orbit phase: After the end of the launch phase at which the launcher will place the satellite into its orbit, the satellite will be floating in space, and hence it becomes a free body, from a mechanical point of view, without any points of support. This means that any outside forces acting on the satellite will lead to it moving as a full body, with no relative motion between its parts. Hence, no significant stresses will develop from this motion, and this phase is the least load-intensive part of the satellite's operational life. However, some small dynamic loading will develop, due to moving parts such as momentum wheels, but their loads will be small compared to those resulting from launch or prelaunch load sources. In addition, some small deflections and resultant thermal stresses will arise due to thermal gradients between the sunlit and non-sunlit parts of the body, but their magnitudes are also small compared to stresses resulting from launch and prelaunch load sources.

Therefore, it is clear that the most critical phase of the satellite's operational life is the launch phase. This phase is usually taken to be the source of critical loads in the design and analysis phases, followed by some portions of the pre-launch phase in secondary importance, especially the transport portions. The in-orbit phase is considered relatively benign in terms of loads. The current work will focus on the launch phase loads.

The subject of load definition is covered by all sources dealing with satellite design, due to its fundamental role in the design and analysis of the structural subsystem. Sarafin, Wijker, and Abdelal et al. [37] all include detailed discussions on the different loads imposed on the structural subsystem during the various phases, starting from pre-launch loads through to the loads resulting from various events that occur during the launch phase. More formalized discussions of loads and load analysis procedures are included in standards and handbooks published by NASA [38] and ESA [39]. It can be seen from reviewing these sources, and others, that the dominant type of loading is dynamic, but with different loading types (e.g., quasi-static, random, shock, etc.) being imposed at different times of the launch phases.

Loads that are imposed on the satellite during one or more phases of its operational life include the following:

- Quasi-static loads: Develop as a result of the response of the inertia of subsystem components and structural components to steady acceleration or slowly varying forces. These loads impose frequencies that are far from the natural frequencies of the components or the system; hence, this type of load will not induce any significant dynamic response. The loads are given in units of $g$ (multiples of the earth's acceleration constant $g$, 9.806 m/s$^2$).
- Random loads: Develop as a result of non-deterministic loading sources, including the mechanical effects of acoustic loads, boundary-layer turbulence mechanical effects, high-frequency engine thrust oscillations, aerodynamic buffeting effects on the launcher fairing, sound pressure effects imposed in the satellite from launcher sources, and others. Random loads are characterized using acceleration power spectral densities (PSD), measured in units of (g$^2$/Hz).
- Shock loads: Develop as a result of very short duration, high-intensity loadings, such as stage separations in multi-stage launchers, and especially the process of separation of the satellite from the launcher at the point of orbital injection. Random loads are characterized by a shock response spectrum (SRS).

In addition to the three types of launch loads outlined above, and which were taken into consideration in the current work, there are other types of loads, such as harmonic, or sinusoidal, and acoustic loads. Based upon the user manual issued by Spaceflight, Inc. [10], which is one of the major launch service providers on a global scale, these loads must be defined after the satellites mounted on any particular launcher are all specified. The launcher authority then computes a coupled-loads analysis (CLA) [39]. This analysis considers the dynamic interactions between the launcher and all satellites mounted upon it, with the results of the CLA including the computation of the values of these loads, and which are then imposed upon the satellites. Since the satellite design for the current work will not be launched, no values for these loads are currently available. As a result, these loads will be disregarded in the current work, and will not be mentioned again. Previous works that considered dynamic launch loads included Aborehab et al. [40], Oh et al. [41], Cote et al. [42], Okuyama et al. [43], Park et al. [44], and others.

## 2. Materials and Methods

This section will provide brief descriptions of the steps that were followed towards developing and implementing a methodology to define the baseline and finalized perforated cases, upon which the loads were imposed. The details of these steps can be found in the previous work [14].

### 2.1. Description of the Baseline Structure's Component Geometry

The structure was designed as a response to a requirement to develop a conceptual satellite designed to carry a remote sensing payload, falling within the microsatellite classification, with a total mass of less than 100 kg. Since no specific launcher was selected at the time the design was developed, two primary requirements during the design and development process were considered. The first was giving the satellite a configuration that would sufficiently fit a suitable imaging system that met the mission requirements. The second requirement was that it be designed to be compatible with multiple launchers, in terms of allowable outer dimensions, considering the allowable static and dynamic envelopes. This multiple launcher compatibility would ensure that the satellite would not be constrained to being launched by only one launcher, hence possibly incurring an unnecessarily high launch cost with no suitable alternatives.

After compiling the necessary dimensional limits for a number of currently active launchers [45–47], it was determined that the maximum dimensions acceptable for multiple launcher compatibility, in terms of the width, height, and depth of a rectangular cuboid-shaped satellite, was one cubic meter. Astronautic design sources cited previously, such as Wertz et al. [33], Sarafin [34], and Wijker [35], state that the most efficient shape for a satellite's structure is that of a rectangular cuboid. This efficiency is due to the best

suitability of fitting the subsystem components within its internal volume in the most efficient manner. This efficiency of cuboid shapes is relative to other possible geometric shapes, such as spherical, cylindrical, or octagonal shapes, as described in works such as that of Aborehab et al. [40], besides being easier to fabricate and assemble.

This was validated by carrying out a number of subsystem component distribution studies, until a satisfactory distribution concept was achieved, as can be seen in Figure 1a. As a result of this process, the satellite's outer width and length were set to equal 0.65 × 0.65 m, and a height of 0.75 m. Based upon past designs, a central load-bearing assembly was added to the structure to act as its backbone and to withstand the highly dynamic loads imposed on the satellite during the launch phase of its operational life. A general view of the geometric shape and outer dimensions of the satellite is shown in Figure 1b.

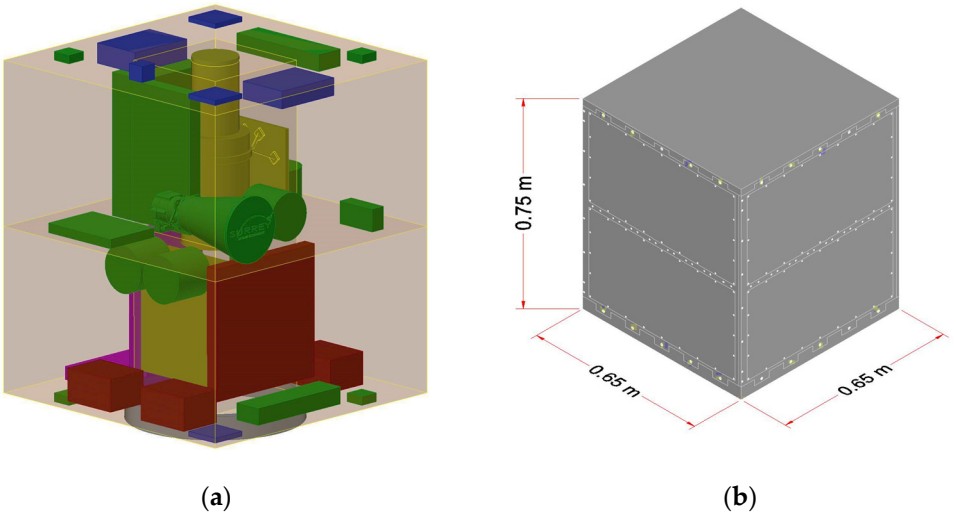

(**a**)          (**b**)

**Figure 1.** (**a**) Microsatellite subsystem configuration concept. (**b**) General outer view and outer dimensions of the microsatellite.

The baseline design for the satellite employed the skin-frame concept, as described by Sarafin. The structure included the following components, as shown in Figure 2:

- A Lightband Mark II separation ring [48], the interface between the satellite and the launch vehicle.
- A set of structural components shown, and listed, in Figure 2. The material for all these components was aluminum 6061 alloy, making this material dominant in terms of the design.
- The structural components listed above were interconnected through hex cap head screws of various lengths. The material selected for all fasteners was titanium 6Al4V. This material was selected for two reasons. The first was that its shear modulus is higher than that for aluminum 6061 alloy, hence leading to a higher resistance to shear. The second was its lower coefficient of thermal expansions relative to that for aluminum 6061, leading to superior thermal creep resistance. Knowing that thermal creep results from the thermal cycling that occurs in orbit, due to the entry and exit of the satellite into and from the shadow of the earth, thermal creep in the fasteners would have resulted in a loosening of the fasteners, which would have led to a reduction of the overall stiffness of the structure.
- To further ensure that the fasteners would not become loose due to vibrations and thermal creep, during both the launch and in-orbit operational phases of the lifetime of the satellite, they were locked into place using Heli-Coil inserts, as described in works by LaRocca et al. [49] and Rainville et al. [50], and others, which were selected based on the sizes defined for each set of fasteners.

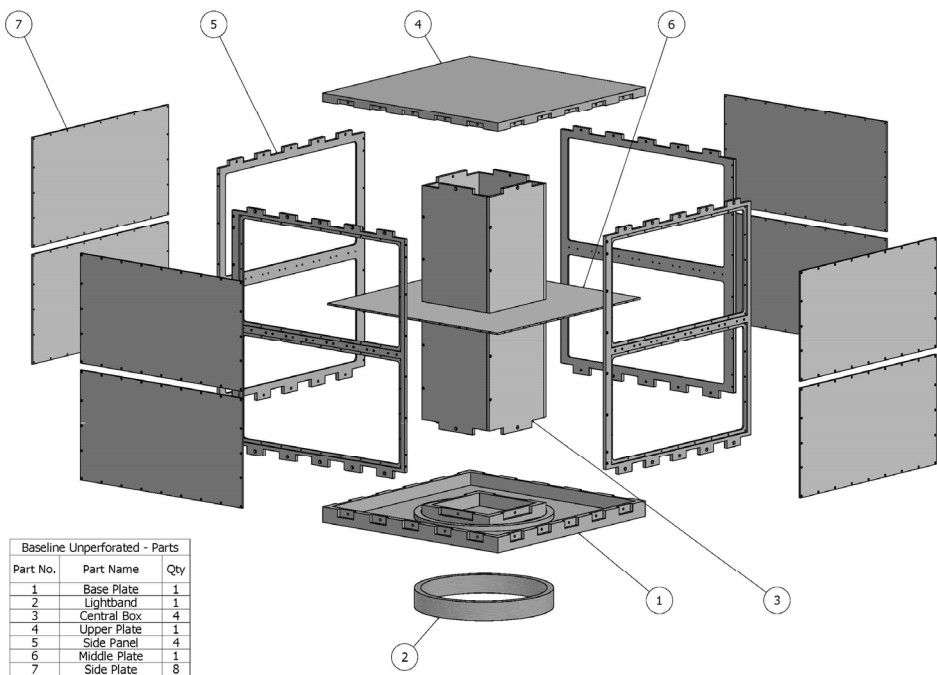

**Figure 2.** Exploded view of the unperforated assembly.

| Baseline Unperforated - Parts | | |
|---|---|---|
| Part No. | Part Name | Qty |
| 1 | Base Plate | 1 |
| 2 | Lightband | 1 |
| 3 | Central Box | 4 |
| 4 | Upper Plate | 1 |
| 5 | Side Panel | 4 |
| 6 | Middle Plate | 1 |
| 7 | Side Plate | 8 |

The three-dimensional modeling software utilized to build and assemble the structure's computational model computed the total mass of the structure as 84.48 kg. This total mass meant that the total mass of the satellite, when fully configured with the components of all the subsystems, would exceed 100 kg, placing the satellite into the minisatellite mass classification, with higher launch costs. Hence, reduction of the structure's total mass was critical to keep the satellite within the sub-100 kg microsatellite classification.

*2.2. Perforation Pattern Implementation*

A set of geometrically defined perforation patterns was developed next. These patterns were implemented onto three structural components, namely each unit of the side plates, the central box plates, and the middle plate. Four geometric pattern design alternatives were developed, namely diamonds, hexagons, squares, and triangles. Each of these four shapes were square-shaped and defined parametrically in terms of the horizontal and vertical dimensions. The thickness of the webs of remaining material from the underlying solid part was also defined parametrically. To explore a broad range of alternatives, a design space was developed through which two aspect ratio and two scale factor alternatives were applied to the patterns that were originally designed, namely 0.5 aspect ratio, 2.0 aspect ratio, half-scale, and double-scale geometric alternatives.

Computational modal analysis was hence implemented, computing the fundamental natural frequency and associated mode shape for each of the design space perforated cases. The unperforated structure was also analyzed and was considered the baseline design. The computational modal analysis methodology was validated through experimental modal analysis procedures.

The results of the computational modal analysis processes led to the selection of one of the design space cases as the best case to be carried forward in the research effort. Namely, the half-scale triangles case was selected, since it was found that this case's fundamental frequency value was the closest to that computed for the baseline case, compared to those values computed for the other perforated cases.

The final step in this stage of the research effort was to apply the tailoring process, to refine the fundamental natural frequency value such that it matched that computed for the baseline case. This process has been employed in many other fields, in works by Chan et al. [51], Tsiatas and Charalampakis [52], Muc [53], Othman et al. [54], and others.

The tailoring process was implemented through changing the number of perforation repetitions that were applied to the central box plates and performing modal analysis on each version of the structure that included a particular repletion. The central box was specifically selected as it was found that changing the unperforated plates to their perforated counterparts had that most influence in increasing the fundamental natural frequency. Including the perforated central box plates had the most influence on the variation of the fundamental natural frequency. A previous work by the authors, and others, explored this influence in detail, as can be seen in [55].

The chosen repetition alternative ensured that implementing the perforation process would not negatively affect the fundamental natural frequency of the structure, while reducing the total mass. This process achieved the final form of the perforated structure, with a percentage reduction in total mass of 23.15%, from 84.48 to 64.92 kg. This final form matched the value of fundamental natural frequency of the baseline case. Figure 3 shows the structural assembly after having reached this finalized form of the perforation patterns.

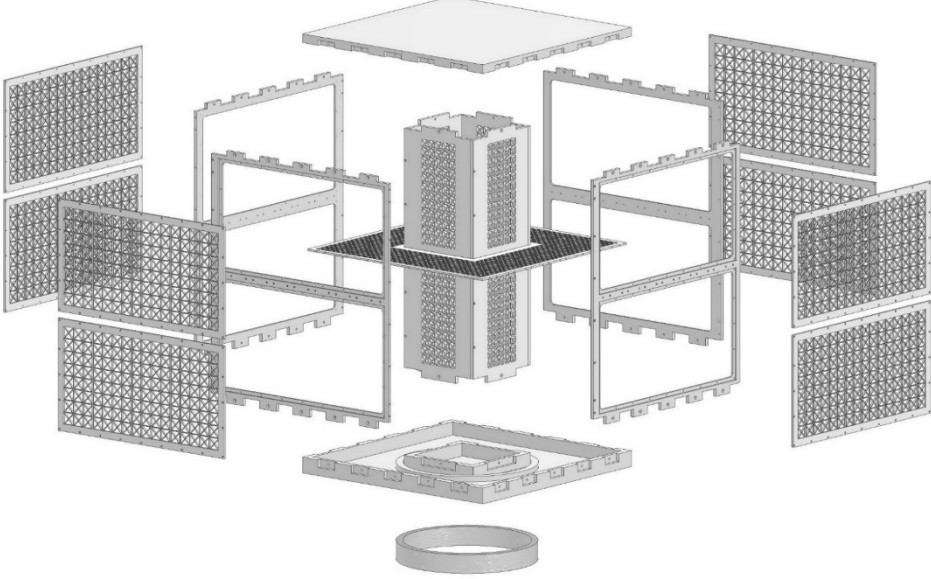

**Figure 3.** Exploded view of the finalized perforated assembly.

### 2.3. Descriptions of the Dynamic Launch Loads

After having achieved the finalized form of the perforated structural assembly in the earlier work, as described above, it was required that the structural performance of this form be studied and compared to the structural performance of the baseline case. This was carried out to investigate the extent to which the presence of the perforations changed the structural responses of the assembly to the expected launch loads. The conclusions from this investigation were considered part of the overall assessment of the concept of implementing the perforation patterns as an alternative to implementing relatively high-cost composite materials in the design.

As mentioned in the introduction, three loading types were considered in the current work, namely, quasi-static, random, and shock loadings. The current work recognizes that other dynamic loading types exist, such as harmonic (sinusoidal), acoustic, and transient loadings. For a space program at a more advanced development stage, these loading types must be considered, to build a complete structural response picture. However, for the current work, it was decided that the three loading types considered would afford a satisfactory picture of the structure's responses to dynamic loadings imposed upon it during the launch phase. Additionally, based upon the Spaceflight manual [56], implementing the other dynamic loading cases would require information from the launch provider. These other loading cases cannot be satisfactorily imposed upon the structure without knowing the structural interactions between the satellite and the other satellites being launched with

it, plus the satellite's interactions with the launcher itself, an analysis process known as a coupled-loads analysis [39]. Since the current satellite was still in the conceptual design phase, and no actual launcher had been selected for it, numerical load values for the other types of loadings could not be determined satisfactorily, and hence these loading cases were disregarded in the current work.

The subsequent sections will present the detailed descriptions of the three dynamic launch loading cases considered in the current work. Section 3 will describe the result sets computed as a result of imposing each of the relevant loading values on the baseline case first, and then on the finalized perforated case. These result sets are in terms of two structural response values specifically, namely the maximum deformations developing in the two cases, as well as the maximum values of equivalent von Mises stresses. Emphasis will be placed upon describing the stress values, because the original satellite design was meant to be implemented in a conceptual earth resources mission, which meant that an imaging payload would be mounted onto the structure. This type of payload would require a structurally stable mounting platform, in terms of ensuring that no permanent structural deformations would develop in the structure as a result of the loading cases being imposed upon it. Thus, it was critical to ensure that the values of von Mises stress developing in the structure would not exceed the structural material's yield stress. Exceeding this value would indicate that plastic deformation had occurred, meaning that permanent deformations would be present in the structure. These permanent deformations would disrupt the carefully calibrated mounting procedures for the imaging payload, which are critical if scientifically meaningful imagery is captured by the satellite. The current work employed the yield margin of safety (MoS) value, introduced below, to quantify whether the von Mises stresses had exceeded the yield stress value of the main structural material or if they had stayed within the material's elastic range, thus indicating a successful structural design.

The current work recognizes the fact that the dynamic loads' analyses described below are based upon nominal loading values provided in the sources that will be mentioned below. Exact loading values can only be given, and applied in analyses, after an actual launcher is selected to launch the satellite into its operational orbit. These exact values can only be computed through performing the aforementioned coupled-loads analyses. As such, the result sets described in the current work must be considered indications of the capacity of the conceptual perforated design to withstand the considered dynamic launch loadings only. More in-depth analyses must be carried out when the design reaches a more advanced state of maturity, and more accurate loading values become available. This recognition is in line with the current state of the design, as described in both the earlier and the current work. The conclusions that will be presented in the subsequent sections below are part of the overall aim of the two works, namely, to verify the possibility of successfully applying perforation patterns to metallic microsatellite structures, to a reasonably high level of confidence.

### 2.3.1. Quasi-Static Loading Analyses

Quasi-static loads are generated during launch due to the nearly constant acceleration of the launch vehicle during its ascent from the launch pad to orbit. In the current work, quasi-static loading values for both longitudinal and lateral directions were quoted from the mission planning guide published by Spaceflight, Inc. (Bellevue, WA, USA) [56], one of the main launch service providers currently in operation. Spaceflight, Inc. act as agents for multiple launcher companies, such as SpaceX (Hawthorne, CA, USA) or Rocket Labs (Long Beach, CA, USA). The selection of a particular launcher is based upon the size of the satellite being launched and the availability of launcher mounting space for that particular satellite.

The source mentioned above specifies a limit load (i.e., the maximum expected load that will be imposed upon the satellite during launch) equal to 10 g in all directions, i.e., 98.1 m/s$^2$, for a microsatellite with a mass less than 100 kg. However, this loading value cannot be

imposed immediately onto the satellite, since a number of design and safety factors must be included to it first.

The ECSS standard concerning structural factors of safety for space hardware (ECSS-E-ST-32-10C Rev.1) [57] provides the values of three factors that were included in the current work. Other factors are also presented in the document, but they were not considered, since they are applicable to more mature phases of the design. The values of these three factors were conservative since various criteria contain a high level of uncertainty. Knowing that the current design is still conceptual, it can be considered relatively immature in terms of the eventual space-worthy system. Table 1 lists the three factors considered in the current work, their values, and brief descriptions of each factor, in terms of their applicability.

**Table 1.** Factors related to dynamic launch loads.

| Factor | Value | Description |
|:---:|:---:|:---:|
| Model Factor ($K_M$) | 1.2 | Accounts for uncertainties in mathematical models predicting dynamic response, loads, and evaluating load paths. |
| Project Factor ($K_P$) | 1.2 | Accounts for how mature a specific project is, in terms of the mass budget, the degree that the design is developed, and the level of confidence in the specifications of the project. |
| Yield Factor of Safety (FOSY) | 1.25 | Accounts for the possibility of material yielding as a result of load application. |

The final loads that were applied to the assembly, in both the longitudinal and lateral directions, were computed through carrying out the following sequence of calculations, as presented in the relevant equations. As mentioned, this sequence was based upon the information provided in the ECSS standard cited above.

1.  The design factor is the product of the multiplication of two factors:

$$DF = K_M \times K_P = 1.2 \times 1.2 = 1.44 \text{ m/s}^2 \qquad (1)$$

2.  The axial and lateral design limit loads, including the calculated design factor (*DF*), were:

$$\text{ADLL} = 98.1 \times 1.44 = 141.264 \text{ m/s}^2 \qquad (2)$$

$$\text{LDLL} = 98.1 \times 1.44 = 141.264 \text{ m/s}^2 \qquad (3)$$

3.  The design yield load was computed through multiplying the design limit loads by the yield factor of safety (FOSY). These loads were applied to the model, and the resulting stresses are known as the design yield stress. The value of FOSY was taken as 1.25 from the ECSS standard above, to afford a higher confidence in the capability of the design to withstand all the loads that it will undergo.

$$\text{Axial Yield Design Load} = \text{ADLL} \times \text{FOSY} = 141.264 \times 1.25 = 176.58 \text{ m/s}^2 \quad (4)$$

$$\text{Lateral Yield Design Load} = \text{LDLL} \times \text{FOSY} = 141.264 \times 1.25 = 176.58 \text{ m/s}^2 \quad (5)$$

An important indication of the capacity of the structural design to survive the application of the dynamic loads is the yield margin of safety (MoS) for the structure. Based upon Beaulieu [58], it can be defined, within the context of the current work, as being the range between two stress levels: the lower limit being a stress value at which the stress could be considered acceptable, in terms of its percentage relative to the material's yield stress, and the higher limit being the material's yield stress value. As can be seen in Equation (6) below, the acceptable stress is the maximum von Mises stress value computed for each case multiplied by the value of FOSY, such that it is lower than the yield stress. The highest value of von Mises stress mentioned was that value that was developed in the aluminum

6061 structural components included in the baseline case, specifically. The reason for this focus on the aluminum 6061 components was due to the fact that this material had the lowest value of yield stress, and hence it was relatively the weakest material in the design. Plus, it also constitutes the material from which the majority of the structural components of the design were fabricated from. The *MoS* was computed in the current work according to the following equation:

$$MoS = \frac{Aluminum\ 6061\ Yield\ Stress}{FOSY \cdot Von\ Mises\ Stress} - 1 \qquad (6)$$

The von Mises equivalent stress values were considered in the current work, instead of other stress values, such as principal stresses. This was because von Mises equivalent stresses can be thought of as representing a form of average stress developing inside a structure, without the need to take any directions into account, hence simplifying the analysis process while preserving an acceptable level of engineering meaningfulness. The process of taking the von Mises stress into consideration, instead of other types of stress, is a widespread practice.

In terms of quantifying the computed values of MoS that will be presented in the subsequent sections, it can be seen from Equation (6) that the value of MoS depends upon the value of the factor of safety (FoS), which is the first part of Equation (6). The factor of safety is defined as a measure of the capability of a system to withstand the expected loads that will be imposed upon it, without failing. In the context of the current work, that would mean the capability of the structural design to withstand the three types of loads that were considered without any material yielding occurring, which was the failure criterion. Hence, the MoS can be seen to be the value of the FoS minus unity. It follows that if the value of the MoS was calculated to be zero, that would mean that the structure can only withstand the exact values of the loads being imposed upon it. As a result, the structure would not have any capacity to withstand any additional loads that might be imposed upon it as a result of inaccuracies in the load value calculations, or an unforeseen loading circumstance. Hence, a positive value of MoS is considered a metric of the success of the design, since that would indicate that the design included a certain amount of reserve capacity, or margin, to withstand any unforeseen loading values.

### 2.3.2. Random Launch Loads' Analysis

As mentioned in the Introduction Section, random vibrations originate in several processes that can occur onboard the launchers as they travel towards orbit while carrying their payloads of satellites. A major source is the process of combustion that occurs inside the rocket engines of the launcher, while another source is the aerodynamic buffeting that is caused by the rapid passage of the launcher through the atmosphere, among other sources.

Random vibrations cannot be characterized by specific, deterministic numbers, being nondeterministic by nature. Hence, they are characterized statistically, through describing power spectral density (PSD) values, across specific frequencies. For the current work, once again, the PSD values implemented in the random vibration analyses were taken from the Spaceflight Inc. user manual. As mentioned before, these values do not represent a specific loading case, such as the limit load utilized in the previous section. They represent a "gateway" that any satellite must successfully pass through before it can be considered for launch onboard a specific launcher. As mentioned before, Spaceflight Inc. offers a number of launcher options, and these gateway, or acceptance, loading values simply allow the company to ensure that a particular satellite can be launched onboard any of the available launchers that it deals with. The acceptance PSD values that were imposed upon both the baseline unperforated and finalized perforated cases, in both the longitudinal ($-Z$) and lateral ($-X$ and $-Y$) directions, equally in all directions, are presented in Table 2. These values of PSD were imposed upon the structure through applying them to vary a 1 g (9.806 m/s$^2$) gravity load in all directions. The random vibration analysis routine within the finite element solver utilized in the current work implemented the mode superposition

analysis technique. This process involves performing a preliminary modal analysis to compute the modes of the system within the frequency range of interest, and these modes were hence utilized to calculate the random vibration response.

**Table 2.** Random vibration loading PSD values.

| Frequency (Hz) | Acceptance PSD (g²/Hz) |
|---|---|
| 20 | 0.056 |
| 40 | 0.056 |
| 50 | 0.06 |
| 800 | 0.06 |
| 1300 | 0.05 |
| 2000 | 0.05 |

2.3.3. Shock Launch Loads' Analysis

A number of sources of shock loads act towards imposing this type of loading upon the satellite during the launch phase of its operational life. One source is the separation of stages that occurs when the first and second stages of the launcher separate. This occurs during the normal sequence of events that occur, while the launch system is ascending from the surface of the earth to the altitude at which its payload of satellites is injected into their orbits.

However, the primary source of shock loading is the separation system that separates the satellite from the launcher at the moment of orbital insertion. The level of this shock loading varies depending on the separation technology being utilized in the separation system. Traditionally, explosive, or pyrotechnic, bolts were utilized to bond the satellite to the launcher while the launcher travelled to its orbital insertion altitude, and hence exploded in unison to separate the satellite at orbital insertion. However, this type of separation system was found to impose a significant level of shock loading upon the satellite. Consequently, newer separation systems utilized motors to separate the ring into two parts at orbital insertion, with one part staying with the launcher, and the other part staying with the satellite during its operational, in-orbit, lifetime. The Lightband II separation ring implemented in the current work belongs to this newer generation of separation systems, utilizing a motor to separate the ring into two halves at orbital insertion.

The user manual of the Lightband II system, as cited previously, includes the chart shown in Figure 4. This figure shows the shock response spectrum of loading values, in multiples of the standard gravitational constant *g* (9.806 m/s²) at a range of frequencies extending from 100 Hz up to a maximum of 10,000 Hz, which is the frequency range of interest in the current work for shock loading.

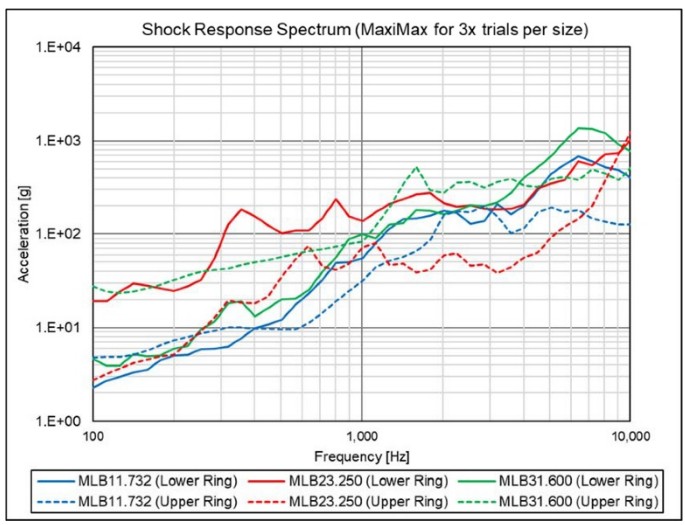

**Figure 4.** Lightband II shock response spectrum loading values [48].

A review of Figure 4 yields two main observations. The first observation is that the shock values are provided at two locations relative to the separation ring: at the upper ring and at the lower ring. In the current work, the values for the lower ring were considered. This was because the separation ring was modeled as part of the structural system, and its response to the shock loading at the moment of separation can be considered the response to the ring as a whole, before it separates into two halves. Hence, the loading values at the lower ring were of interest since the lower ring was considered the boundary between the structural system and the outside world. This consideration was also reflected in the process of clamping the ring's lower face.

The second observation from Figure 4 was that the values of acceleration, presented in g factors, were provided for only three values of diameter for the ring, namely 11.732, 23.25, and 31.6 inches. Returning to the original design of the structure, as given in the previous work, it can be found that the geometric dimensions of the base plate resulted in the selection of a suitable diameter of the Lightband II ring, namely the 15-inch diameter variant. Hence, a process of interpolation was required to estimate the g factors that are imposed upon the structure when the 15-inch model was implemented in the current work. This interpolation process was achieved through implementing Equation (7), derived from linear interpolation principles, knowing that the values of frequency for both the 11.732- and 23.25-inch models were the same.

$$\frac{D_{23.25} - D_{15}}{D_{23.25} - D_{11.732}} = \frac{g_{23.25} - g_{15}}{g_{23.25} - g_{11.732}} \tag{7}$$

Filling in the known values of the diameters of each model, in terms of the diameters of each model and the corresponding g factor values for the 11.732- and 23.25-inch models, as estimated from Figure 4, with as much accuracy as possible, the acceleration values for the 15-inch model were hence computed. The computed values of acceleration for the 15-inch separation ring model, both in terms of g factors and in terms of acceleration values, as required by ANSYS, were found through multiplying the g factor values by 9.806 m/s$^2$. The g factor values for the three models, plus the acceleration values for the 15-inch model, are presented in Table 3. These values were imposed upon the computational models of the baseline unperforated and finalized perforated structures.

**Table 3.** Shock response spectrum for 15-inch diameter Lightband II.

| Frequency (Hz) | Acceleration (g) (11.732-inch) | Acceleration (g) (23.25-inch) | Acceleration (g) (15-inch) (Interpolated) | Acceleration (m/s$^2$) (15-inch) |
|---|---|---|---|---|
| 100 | 2.3 | 11 | 4.771 | 46.783 |
| 200 | 5 | 11.500 | 6.846 | 67.132 |
| 300 | 6 | 80 | 27.016 | 264.919 |
| 400 | 10 | 160 | 52.600 | 515.796 |
| 500 | 13 | 100 | 37.708 | 369.765 |
| 600 | 20 | 110 | 45.560 | 446.761 |
| 700 | 30 | 145 | 62.660 | 614.444 |
| 800 | 50 | 240 | 103.960 | 1019.432 |
| 900 | 50 | 160 | 81.240 | 796.639 |
| 1000 | 55 | 150 | 81.980 | 803.896 |
| 2000 | 180 | 210 | 188.520 | 1848.627 |
| 3000 | 180 | 190 | 182.840 | 1792.929 |
| 4000 | 200 | 205 | 201.420 | 1975.125 |
| 5000 | 450 | 350 | 421.600 | 4134.210 |
| 6000 | 600 | 480 | 565.920 | 5549.412 |
| 7000 | 700 | 580 | 665.920 | 6530.012 |
| 8000 | 500 | 700 | 556.800 | 5459.981 |
| 9000 | 500 | 750 | 571.000 | 5599.226 |
| 10,000 | 400 | 1000 | 570.400 | 5593.342 |

*2.4. Methodology for Dynamic Loads' Computational Analysis*

The following chronological steps were developed as the methodology for computing the structural responses of the analyzed cases to the dynamic loads in the current work. The finite element analysis computational method was employed to generate the result sets presented in Section 3, and the ANSYS solver was utilized, since it offered a number of results' postprocessing features that will be highlighted in Section 3, compared to other finite element analysis systems.

1.  Solid models of the structural components, previously built using 3D modeling software, were imported into a new ANSYS Workbench project, within the Geometry ANSYS system.
2.  The ANSYS Geometry system data were then shared to two ANSYS modal analysis systems. The first analysis system was set to compute modal characteristics up to a frequency of 2000 Hz. The modal results from this analysis were fed into the subsequent random analysis stages, and the second analysis system was set to compute modal characteristics up to a frequency of 10,000 Hz, to feed its results to the subsequent response spectrum stages. The ANSYS modal analysis systems were required because both the random and shock analyses depend upon precomputed modal results, as part of the modal superposition analysis processes that ANSYS implements.
3.  As such, three ANSYS random analysis systems were derived from the first modal analysis system, one for each of the three directions of interest (along the X, Y, and Z directions), representing the two lateral and the longitudinal directions, respectively. Additionally, three ANSYS response spectrum analysis (for the shock analyses) systems were derived from the second modal analysis system, one for each direction. The first modal system was set to compute all vibrational modes falling within the frequency range of interest for the random analyses. This was based upon the nominal input loads presented in the preceding section. The same process was performed for the second modal system, to include the relevant frequency range for the shock analyses.
4.  Additionally, from the initial Geometry system, three new static analysis systems were derived, one for each direction, as mentioned above, within which the quasi-static analyses were defined, after having defined the material properties within it.
5.  Surface contacts were automatically defined in all the modules mentioned above, between all the components, a feature of the Workbench Mechanical application. Bonded contacts between all the structural components and interconnecting fasteners were implemented, to model the fact that implementing the fasteners resulted in a structural assembly with no allowable relative motion between any components.
6.  The structural components were discretized into finite elements through the meshing functionality. The specific type of finite element used to carry out all analyses was the 10-noded, higher order, tetrahedral element known as SOLID186, as included within the ANSYS element library [59]. This element type allows a lower number of elements to be used to discretize the structural components than what would have been necessary if lower-order elements had been used.
7.  The final step in the dynamic loads' analysis methodology, across all the analysis types relevant to the current work, was to impose boundary conditions upon each model. These boundary conditions accurately reflected the physical constraints applied to the satellite while inside the launcher. As such, to model the fact that the Lightband II separation ring is connected to the launcher on its lower face, a fixed boundary condition was applied to a point defined to be in the center of the lower face of the separation ring, and which was connected to the inner edge of the lower face by rigid links. This method of support was required by the random and shock analysis systems.

As mentioned in the previous work, mesh convergence was employed in the current work to develop finite element meshes that would produce results with a reasonably

acceptable degree of accuracy. This accuracy was meant to be compared to results that would have been acquired had physically fabricated structural components of the cases been produced and tested. The convergence studies involved creating meshes within the ANSYS preprocessor that included element sizes that were automatically computed by the preprocessor, based upon the dimensions of each part. After running an analysis employing this initial mesh, and recording the particular result of interest, the element sizes were reduced systematically, and the model was reanalyzed. This process was repeated until it was seen that the result of interest did not change appreciably from one element size to the next. Hence, the element size was considered to have converged, and the result computed by the earlier element size was considered the definitive result for that particular case.

## 3. Results

The generation of result sets in the current work included a two-step process. Firstly, dynamic loading analyses, for each of the three types of loads presented in the Introduction Section, were performed for the baseline case, to produce the baseline results. Care was taken to observe that the computed yield margin of safety for each case was positive in value. Secondly, the same set of analyses were performed for the finalized perforated case, and the result sets computed for each of these analyses were compared to the baseline result sets. These comparisons led to the conclusions drawn in the current work, and contributed to the overall conclusion, pursued by both the previous work and the current work, regarding the validity and benefits of implementing the perforation patterns' mass reduction concept. As such, the following subsections will present both result sets to make comparisons between the two, and conclusions for each case, easier.

### 3.1. Launch Loads' Analyses of the Perforated vs. Baseline Cases

After achieving the final definitive form of the perforated satellite structural assembly design in the previous work, this design was subjected to the same dynamic loads that were imposed upon the baseline case. This was carried out for analyzing its responses and comparing them to those computed for the baseline case. These comparisons provided the final indication of the validity of the current work's proposed approach to mass reduction. Namely, if the results computed for the three dynamic launch loads proved that the finalized perforated design did have the capacity to withstand these loads without undergoing material yield failure.

### 3.1.1. Quasi-Static Loading Analysis Results

Table 4 presents a summary of the quasi-static loading analysis results, across both the baseline results provided before plus the results computed for the finalized perforated case, in terms of the maximum von Mises stress results. It also shows the components that developed these values, in addition to the computed yield MoS for each case.

The maximum deformation result, across all loading directions, was computed to be $3.766 \times 10^{-4}$ m, computed for the lateral loading direction. As before, this value was seen to be very low, due to the relatively high system stiffness. Another observation was that the mentioned value was less than that computed for the baseline case, due to the reduced total mass of the structure as a result of the implementation of the perforations. This reduction led to a reduction in the inertial loadings developing within the structure, and hence to a reduction in the resulting deformations.

In terms of the maximum von Mises stress result sets in Table 4, two main observations were noted. The first observation regarded the results of the analyses considering the two lateral directions. In these two cases, it was seen that the maximum von Mises stress values developed in the base plate, for both the baseline and finalized perforated cases. This was seen to be a logical consequence of the deformation patterns mentioned above. Namely, in these types of deformations, the maximum stresses develop in the parts or regions closest to the clamping locations. This occurs because the inertial loads that developed as a consequence of imposing the quasi-static loads act at their maximum levels

at these clamping locations. This was because they are clamped, meaning that little or no possibilities of relative movement exist at these locations, and the inertial loads cannot be dissipated, hence they act at their maximum levels. It was seen that the values of von Mises stress were lower for the finalized perforated case than the values for the baseline case. This was attributed to the fact that the mass of the structural assembly was significantly less for the finalized perforated case, as was the main objective of the current work. This meant that the von Mises stresses developing as a result of the inertial loads were lower, because these loads were less, since inertial loads vary in direct proportion to the masses that move and generate these loads. As a consequence of the reduced von Mises stresses that developed for the finalized perforated case, the computed yield MoS values were higher, even though these values were positive for all cases. This indicated that including the perforations developed in the current work had a beneficial effect on the structure's capacity to withstand quasi-static loads in the lateral directions.

**Table 4.** Baseline and perforated quasi-static launch loads—maximum von Mises stresses.

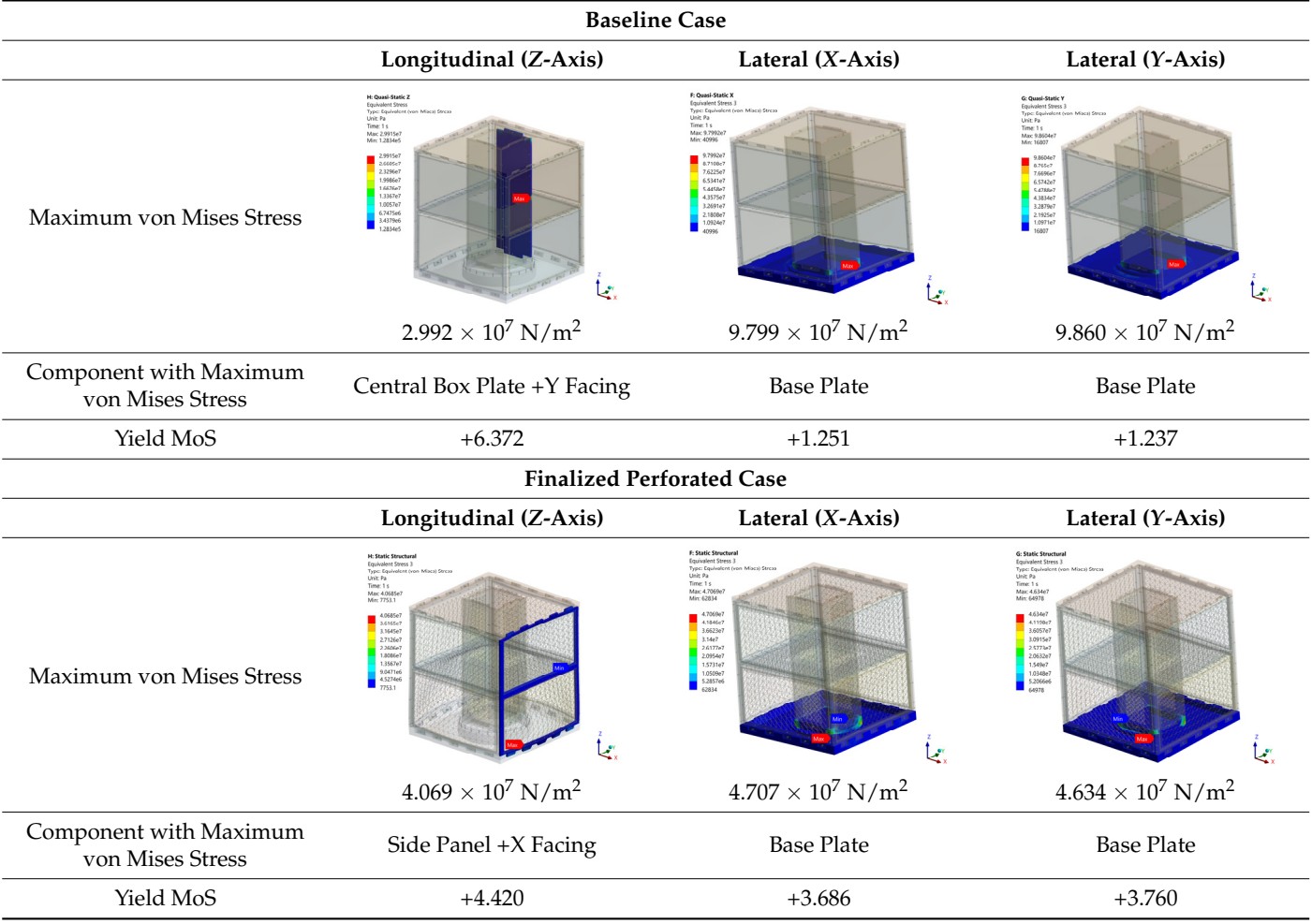

| | **Baseline Case** | | |
|---|---|---|---|
| | **Longitudinal (Z-Axis)** | **Lateral (X-Axis)** | **Lateral (Y-Axis)** |
| Maximum von Mises Stress | $2.992 \times 10^7$ N/m$^2$ | $9.799 \times 10^7$ N/m$^2$ | $9.860 \times 10^7$ N/m$^2$ |
| Component with Maximum von Mises Stress | Central Box Plate +Y Facing | Base Plate | Base Plate |
| Yield MoS | +6.372 | +1.251 | +1.237 |
| | **Finalized Perforated Case** | | |
| | **Longitudinal (Z-Axis)** | **Lateral (X-Axis)** | **Lateral (Y-Axis)** |
| Maximum von Mises Stress | $4.069 \times 10^7$ N/m$^2$ | $4.707 \times 10^7$ N/m$^2$ | $4.634 \times 10^7$ N/m$^2$ |
| Component with Maximum von Mises Stress | Side Panel +X Facing | Base Plate | Base Plate |
| Yield MoS | +4.420 | +3.686 | +3.760 |

The second main observation that was made from reviewing the maximum von Mises stress result sets in Table 4 was related to the longitudinal direction. This observation was not as clear-cut as was the case for the analyses for the lateral directions. The reason for this was that the structural component that developed the maximum von Mises stress was not the same across the baseline and perforated cases. To aid the investigation of the von Mises stresses, the von Mises stress results for the longitudinal direction, for both cases, were focused upon in Table 5.

**Table 5.** Baseline and perforated quasi-static launch loads—von Mises stresses, longitudinal direction.

| Baseline Case | | |
|---|---|---|
| | **Longitudinal (Z-Axis)** | **Longitudinal (Z-Axis)** |
| von Mises Stress |  $2.992 \times 10^7$ N/m² |  $8.613 \times 10^6$ N/m² |
| Component with von Mises Stress | +Y Facing Central Box Plate | +X Facing Side Panel |
| Yield MoS | +6.372 | +24.608 |
| **Finalized Perforated Case** | | |
| | **Longitudinal (Z-Axis)** | **Longitudinal (Z-Axis)** |
| von Mises Stress |  $6.140 \times 10^6$ N/m² |  $4.069 \times 10^7$ N/m² |
| Component with von Mises Stress | +Y Facing Central Box Plate | +X Facing Side Panel |
| Yield MoS | +34.922 | +4.420 |

Focusing first on the central box plate, it was seen that value of von Mises stress was significantly reduced for the finalized perforated case, relative to the baseline case, with an associated increase in yield margin of safety. The following deduction was reached to explain this variation in values. A relative reduction in stiffness distribution for this part had occurred as a result of the perforation process. The quasi-static loading values can be seen to have units of acceleration, as presented in Table 3. Therefore, when these loadings were imposed upon the structure, they generated inertial loads inside the components. As a result, these inertial loads generated the displacements and von Mises stress values provided earlier for the baseline case. However, due to the reduced mass of the plate, as a result of the perforation process, the value of induced inertial load on the plate, as a result of the quasi-static loading, was also reduced, since inertial loading is proportional to mass. The von Mises stress was reduced in the perforated central box plate, relative to the baseline plate, thereby increasing the yield margin of safety. This indicated that the effect of the inertial load, even though it was reduced from its value in the baseline case, had a larger part to play than the reduced stiffness distribution, which would have increased the stress values due to the relative weakening of the plates. However, it was also observed that the yield margin of safety still remained positive. This meant that, despite the relative weaking of the part, it was still viable as a load-bearing component and did not lead to it being a source of structural failure for the structure as a whole.

Focusing on the side panel, it was seen that the significant increase in von Mises stresses in the finalized perforated case, relative to the baseline case, was deduced to have been caused by a reverse of the process described for the central box plate. An associated significant decrease in yield margin of safety occurred in this case. This decrease occurred even though the masses of the perforated middle plate and side plates were reduced, relative to those for the baseline case. As a result of the perforations, the stiffness

distributions of the perforated middle plate and side plates had larger parts to play. These stiffness distributions thereby weakened the parts, even though they reduced as a result of the perforation process, hence increasing the effect of the inertial load, thus increasing the von Mises stress developing in the side panel.

However, as mentioned above for the central box plate, it was also seen that the value of yield margin of safety, even though it was reduced, still stayed positive. This led to the deduction that the relative weakening of the structure as a result of the presence of the perforations did not lead to any yield failures developing in its components. Nevertheless, the relatively high values of MoS also indicated a potential for further mass reductions, as mentioned before.

In the final analysis, it was seen that the stiffness distribution reductions in the perforated components did not have a large detrimental effect on the system's overall structural performance. This was because the MoS values for the finalized perforated case were all positive, in all directions, indicating that the reduced overall stiffness distribution of the system was not significant enough to lead to yield failure in the structure.

### 3.1.2. Random Loading Analysis Results

Table 6 presents a comparison between the maximum von Mises stress values across the baseline case, as mentioned before, and the finalized perforated case, for all three directions, as shown before. The values in Table 6 are also 3σ values, due to the same aforementioned reason, namely the fact that the loading values of Table 2 are non-deterministic, and hence the computed maximum von Mises stress values are also statistical in nature. As mentioned before, the fact that they are 3σ values means that for 99.73% of the analysis time, the computed values are equal to or less than the maximum values presented in Table 6.

The maximum deformation result, across all loading directions, was computed to be $1.398 \times 10^{-4}$ m, computed for the lateral loading direction. As mentioned before, this value was considered very low, due to the relatively high system stiffness. The given value was also less than that computed for the baseline case, due to the same deductions mentioned in the previous section.

In terms of the computed 3σ values of maximum von Mises stress, as shown in Table 6, a repeat of the observations made for the quasi-static loading analyses in the longitudinal, *Z*-axis direction was seen. This also led to Table 7, which contained the same results as Table 5, arranged in the same manner.

As can be seen from Table 7, it can be observed that, similar to the quasi-static loading case, the maximum von Mises stress values were developed in different structural components across the two cases. A deduction process similar to the one employed to investigate the results in Table 5 was employed to analyze the variations in stress results in Table 7. This deduction process arrived at similar conclusions as for the quasi-static loading analyses. Namely, the relation between the induced inertial loads, as a result of the imposed loadings, and the relative weakening effects of the presence of the perforations. However, it was noted that the values of the yield margin of safety for the finalized perforated case were still positive, as seen in the quasi-static case. Once again, this led to the deduction that the presence of the perforations did not have a large enough detrimental effect to cause material yielding in the structure's components.

In terms of the lateral, *X*-axis and *Y*-axis, directions, it was seen that a significant drop in maximum von Mises stress values developed, between the baseline case and the perforated case, and in the same structural component, namely the base plate, for all cases. This was also a repetition of what was observed in the quasi-static analysis cases, but with a larger drop in maximum von Mises stress levels for the finalized perforated cases. This indicated a higher sensitivity to the reduced system stiffness in the random analysis cases, relative to the quasi-static analysis cases.

Again, in the final analysis, it was seen that all computed values of MoS were positive, across all directions, for both baseline and finalized perforation cases. This led to the conclusion that the structure would successfully withstand random loading, with a reasonable

amount of confidence, even with the relatively reduced system stiffness resulting from the inclusion of the perforations. It was also deduced, based upon the relatively high values of MoS, that there is a potential to further reduce the mass.

**Table 6.** Baseline and perforated random launch loads—maximum von Mises stresses.

| Baseline Case | | | |
|---|---|---|---|
| | **Longitudinal (Z-Axis)** | **Lateral (X-Axis)** | **Lateral (Y-Axis)** |
| Maximum von Mises Stress (3σ Values) | 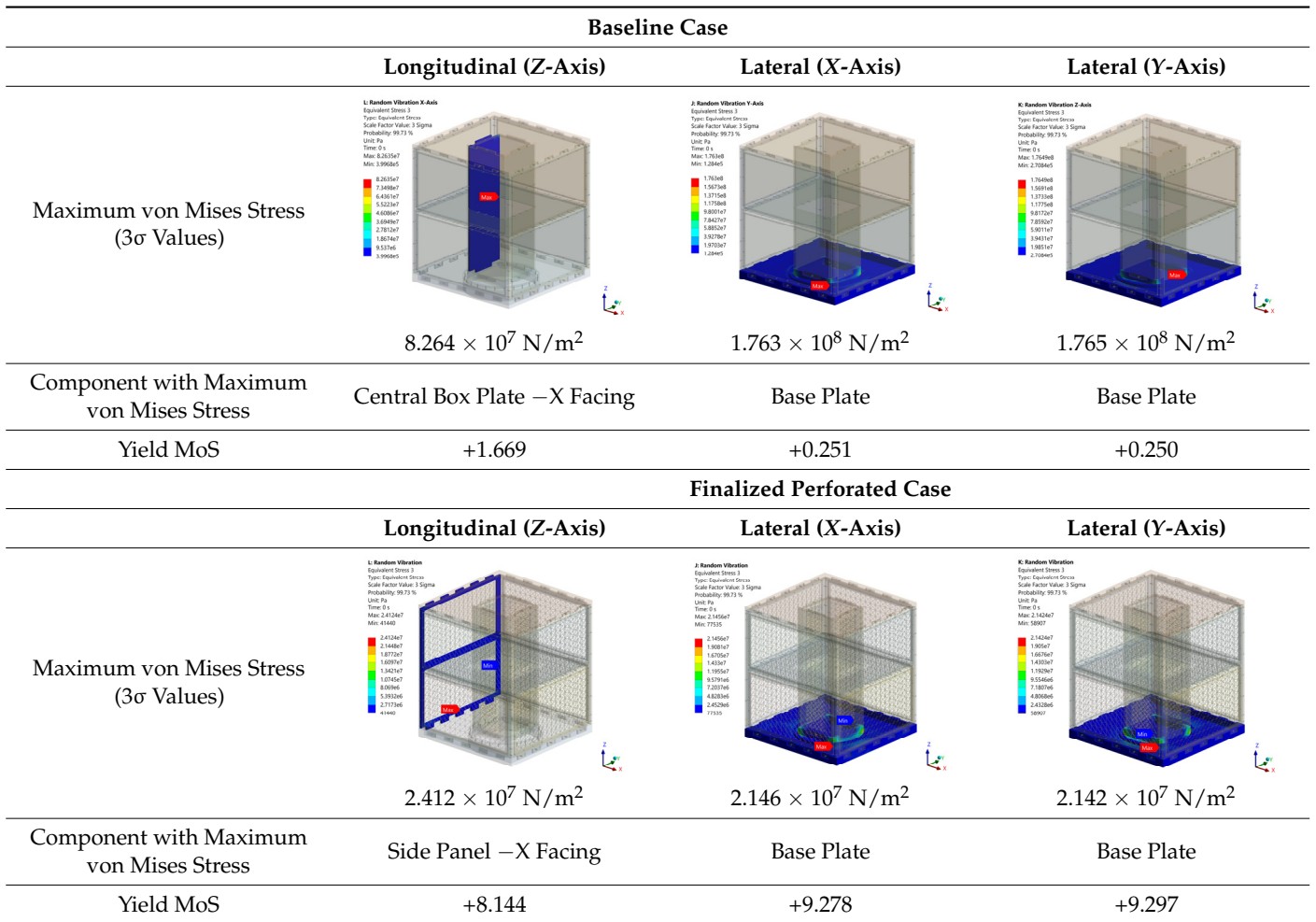 | | |
| | $8.264 \times 10^7$ N/m$^2$ | $1.763 \times 10^8$ N/m$^2$ | $1.765 \times 10^8$ N/m$^2$ |
| Component with Maximum von Mises Stress | Central Box Plate $-$X Facing | Base Plate | Base Plate |
| Yield MoS | +1.669 | +0.251 | +0.250 |
| **Finalized Perforated Case** | | | |
| | **Longitudinal (Z-Axis)** | **Lateral (X-Axis)** | **Lateral (Y-Axis)** |
| Maximum von Mises Stress (3σ Values) | | | |
| | $2.412 \times 10^7$ N/m$^2$ | $2.146 \times 10^7$ N/m$^2$ | $2.142 \times 10^7$ N/m$^2$ |
| Component with Maximum von Mises Stress | Side Panel $-$X Facing | Base Plate | Base Plate |
| Yield MoS | +8.144 | +9.278 | +9.297 |

### 3.1.3. Shock Loading Analysis Results

Table 8 presents the computed maximum von Mises results for both the baseline case, as presented previously, and for the finalized perforation case. These results were computed after imposing the shock loadings originating from the operation of the Lightband II separation ring, as in Table 4. In both tables, the results are presented for the longitudinal, Z-axis, direction, and for the lateral, X-axis and Y-axis, directions, as presented in the preceding sections. These results are deterministic, and are actual, specific result values, similar to the results presented for the quasi-static analyses.

The maximum deformation result, across all loading directions, was computed to be $1.485 \times 10^{-4}$ m, computed for the lateral loading direction. Again, this value was very low, with the relatively high system stiffness. However, in this case, it was seen that the maximum deformation value for the perforated cases was higher than that for the baseline case, and this was attributed to the stiffness distribution of the perforated case, which led to the increased deformations. Nevertheless, the maximum value was still very low, as mentioned.

Reviewing the maximum von Mises stress values in Table 8, that developed due to the imposition of the shock loading in Table 4, for both the baseline case and the finalized perforated case led to several observations. The first observation was that, yet again, the same variation of stresses was seen between the baseline case and the finalized perforated

case, for the longitudinal, *Z*-axis, direction. Namely, it was seen that the component that developed the maximum value for the baseline case was the base plate, whereas for the finalized perforated case, it was the middle plate. Table 9 presents the same results' data regarding the variations in von Mises stresses as those presented in both Tables 5 and 7.

**Table 7.** Baseline and perforated random launch loads—von Mises stresses, longitudinal direction.

| | Baseline Case | |
|---|---|---|
| | **Longitudinal (*Z*-Axis)** | **Longitudinal (*Z*-Axis)** |
| von Mises Stress (3σ Values) |  $8.264 \times 10^7$ N/m$^2$ |  $1.811 \times 10^7$ N/m$^2$ |
| Component with von Mises Stress | Central Box Plate −X Facing | Side Panel −X Facing |
| Yield MoS | +1.669 | +35.529 |
| | **Finalized Perforated Case** | |
| | **Longitudinal (*Z*-Axis)** | **Longitudinal (*Z*-Axis)** |
| von Mises Stress (3σ Values) |  $2.896 \times 10^6$ N/m$^2$ |  $2.412 \times 10^7$ N/m$^2$ |
| Component with von Mises Stress | Side Panel −X Facing | Central Box Plate −X Facing |
| Yield MoS | +75.160 | +8.144 |

**Table 8.** Baseline and perforated shock launch loads—maximum von Mises stresses.

| | Baseline Case | | |
|---|---|---|---|
| | **Longitudinal (*Z*-Axis)** | **Lateral (*X*-Axis)** | **Lateral (*Y*-Axis)** |
| Maximum von Mises Stress |  $2.677 \times 10^7$ N/m$^2$ |  $3.217 \times 10^7$ N/m$^2$ |  $3.119 \times 10^7$ N/m$^2$ |
| Component with Maximum von Mises Stress | Base Plate | Base Plate | Base Plate |
| Yield MoS | +7.239 | +5.865 | +6.071 |

**Table 8.** *Cont.*

| Finalized Perforated Case | | |
|---|---|---|
| **Longitudinal (Z-Axis)** | **Lateral (X-Axis)** | **Lateral (Y-Axis)** |
| 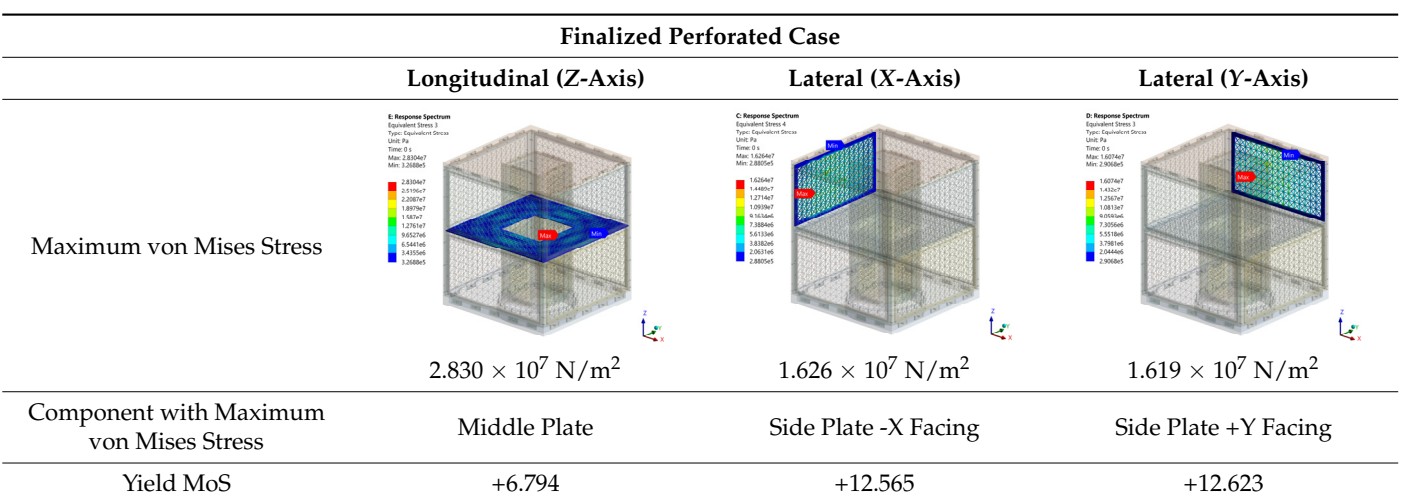 | | |
| $2.830 \times 10^7$ N/m$^2$ | $1.626 \times 10^7$ N/m$^2$ | $1.619 \times 10^7$ N/m$^2$ |

| | Finalized Perforated Case | | |
|---|---|---|---|
| Maximum von Mises Stress | | | |
| Component with Maximum von Mises Stress | Middle Plate | Side Plate -X Facing | Side Plate +Y Facing |
| Yield MoS | +6.794 | +12.565 | +12.623 |

**Table 9.** Baseline and perforated shock launch loads—von Mises stresses, longitudinal direction.

| Baseline Case | |
|---|---|
| **Longitudinal (Z-Axis)** | **Longitudinal (Z-Axis)** |
| $2.677 \times 10^7$ N/m$^2$ | $1.498 \times 10^7$ N/m$^2$ |

| | Baseline Case | |
|---|---|---|
| von Mises Stress | | |
| Component with von Mises Stress | Base Plate | Middle Plate |
| Yield MoS | +7.239 | +13.724 |

| Finalized Perforated Case | |
|---|---|
| **Longitudinal (Z-Axis)** | **Longitudinal (Z-Axis)** |
| $1.982 \times 10^7$ N/m$^2$ | $2.830 \times 10^7$ N/m$^2$ |

| | Finalized Perforated Case | |
|---|---|---|
| von Mises Stress | | |
| Component with von Mises Stress | Base Plate | Middle Plate |
| Yield MoS | +10.128 | +6.794 |

A review of the results in Table 9 yielded observations that were similar to those noted for the detailed von Mises stress analyses between the perforated results and the baseline results, for both the quasi-static and random loadings. Namely, for the base plate, the presence of the perforations had reduced the mass of the structure. This reduced the magnitude of the inertial load resulting from it, and the reduced inertial load being imposed upon the base plate led to a reduced value of von Mises stress to develop in it, thereby increasing the yield margin of safety.

Similar to what was observed before, regarding the middle plate, even though the perforations caused a reduction in the stiffness distribution, the effect of this reduction was lower than the effect of the reduction in inertial load resulting from the reduction in mass. As a result, a larger value of von Mises stress developed in the perforated middle plate than for the baseline middle plate, thereby reducing the yield margin of safety from that computed for the baseline case. It was noted that the change in yield margins of safety was not as significant as those noted for the quasi-static and random cases, indicating that the reduction in overall stiffness was not as sensitive to the presence of the perforations as in the other two loading cases.

For the lateral, *X*-axis and *Y*-axis, directions, the values of maximum von Mises stress, as can be seen in Table 8, saw a reduction for the finalized perforated case, relative to the baseline case. Additionally, yet again, the components that developed the maximum values of von Mises stress changed from the base plate, for the baseline case, in both directions, to one of the side plates in the both the *X*-axis and *Y*-axis cases. As mentioned above, the deduction was that the effect of the reduction in the stiffness distribution of the perforated middle plate was less than that of the reduction of the inertial load. This led to an increase in the value of von Mises stress, and a reduction in the yield margin of safety.

However, as mentioned before in both the quasi-static and random loading cases, it was observed that the reduction in stiffness due to the implementation of the perforations did not drastically alter the structure's capacity to withstand the shock loads, since the values of MoS were all positive. The positive MoS values indicated that the structure had developed stresses that were still within the material's yield stress limit after application of the shock loading. Additionally, the relatively high values of MoS for the perforated cases indicated the potential for further mass reduction.

## 4. Discussion and Conclusions

Having presented the current work's various result sets in the previous section, it can be definitively concluded that the implementation of the perforation pattern concept was viable as a low-cost alternative to implementing composite materials for the reduction of mass in structural subsystems of microsatellite and higher classed small satellites. This implementation process was described in detail in both the previous and current works.

To be more specific, in terms of the mass percentage reduced, relative to the baseline, unperforated case, a final reduction percentage of 23.15% was seen to have been significant. Additionally, in terms of the structural responses to three of the dynamic launch loadings that would be imposed upon the system while it endured the launch phase of its operational life, the presence of the perforations, due to their mass-reducing effect, was numerically proven to have had a beneficial effect on these responses.

In terms of recommendations for moving the overall research work forward in future works, it is suggested that formal, multi-objective mass optimization procedures be implemented on the finalized perforated design. This was towards removing an additional amount of material from the perforated structural components, through utilizing the MoS values as metrics, in addition to the previous work's focus on the fundamental natural frequency. In addition, comparative studies can be performed implementing the same structural design, but involving composite materials instead of the perforation patterns, to further study the viability of implementing and developing the perforation pattern concept in the future.

**Author Contributions:** Research investigation and draft preparation, S.D.S.D.; project administration, funding acquisition, manuscript review and supervision, M.Y.H. All authors have read and agreed to the published version of the manuscript.

**Funding:** This research work was funded by Universiti Putra Malaysia (UPM), Selangor, Malaysia, through research grant number GP-IPS-9536500.

**Institutional Review Board Statement:** Not applicable.

**Informed Consent Statement:** Not applicable.

**Data Availability Statement:** Not applicable.

**Acknowledgments:** The authors acknowledge the support from Universiti Putra Malaysia (UPM), Selangor, Malaysia, on the research effort for the current work through research grant GP-IPS-9536500.

**Conflicts of Interest:** The authors declare no conflict of interest.

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
