# Peer review of "Structural Responses of a Conceptual Microsatellite Structure Incorporating Perforation Patterns to Dynamic Launch Loads"

_aerospace, doi:10.3390/aerospace9080448_

Round 1

Reviewer 1 Report

The manuscript is well written except for a few minor points regarding the presentation.

The previous work done by the authors regarding perforation pattern-based mass reduction is heavily relied upon. Authors should reduce the amount of information being repeated here. For example, Section 2 has too much information coming from previous design work. Authors should reduce the overlap significantly and only very important and relevant information needs to be compiled as a summary in this work. For example, the material and design parameters are the only relevant part from previous work that is needed here.

The sentences are too long. For example, in line 255, 'Astronautic design sources cited previously, such as Wertz et al, Sarafin, and Wijker, state that the most efficient shape for a satellite’s structure is that of a rectangular cuboid due to the best suitability of fitting the subsystem components within its internal volume in the most efficient manner, relative to other possible shapes, such as spherical or cylindrical shapes, as described in works such as Aborehab et al. [40], compared to other geometric shapes, e.g., cylindrical, hexagonal, octagonal, and others, besides being easier to fabricate and assemble.'

There is a similar structure in many places. This makes it difficult to process information.

Reviewer 2 Report

1. In section 2.1, how did the authors determine the outer dimensions of the satellite?

2.The material of the satellite component is aluminum 6061 alloy, while the material selected for the fastener is titanium 6Al4V. Why did the author not use the same material?

3.In the implementation of modal analysis in section 2.3 of the article, how did the author obtain the half-scale triangle case as the best case?

4.The article mentions that the authors' proposed method reduces mass by 23.15% relative to the baseline, non-perforated case. This percentage reduction is so significant that the authors can confirm it again.

Reviewer 3 Report

The work investigates the dynamic response characteristics of a perforated satellite. The layout of the study is well done and the objectives are clear. The structure under investigation is explained, and the nature of the launch loads are described. Computations are undertaken using Ansys, and the details of the FE model are well explained. The results are verified based on the results of past work, and the findings are useful for designers and analysts involved in the design of such structures.

Some minor corrections include:

The notation for the model and project factor should include the subscript to be consistent with Equation (1).

Equation numbers (4) and (5) should be right justified.
